# Novel Synthetic Steroid Derivatives: Target Prediction and Biological Evaluation of Antiandrogenic Activity

**DOI:** 10.3390/cimb47121059

**Published:** 2025-12-17

**Authors:** David Calderón Guzmán, Norma Osnaya Brizuela, Hugo Juárez Olguín, Maribel Ortiz Herrera, Armando Valenzuela Peraza, Ernestina Hernández Garcia, Alejandra Chávez Riveros, Sarai Calderón Morales, Alberto Rojas Ochoa, Aylin Silva Ortiz, Rebeca Santes Palacios, Víctor Manuel Dorado Gonzalez, Diego García Ortega

**Affiliations:** 1Laboratory of Neurosciences, Instituto Nacional de Pediatria (INP), Mexico City 04530, Mexico; davidcaguama91@gmail.com (D.C.G.); osnayanorma@hotmail.com (N.O.B.); valenzuela.peraza2013@hotmail.com (A.V.P.); diegortega.qfb@gmail.com (D.G.O.); 2Laboratory of Pharmacology, Instituto Nacional de Pediatria (INP), Mexico City 04530, Mexico; ernestinahg@prodigy.net.mx; 3Laboratory of Experimental Bacteriology, Instituto Nacional de Pediatria (INP), Mexico City 04530, Mexico; mortizherrera@hotmail.com; 4Laboratory of Pharmacy, Faculty of Chemist, UNAM, Mexico City 04530, Mexico; achavezr@hotmail.com (A.C.R.); asilvao@hotmail.com (A.S.O.); 5Experimental Biology, Universidad Autónoma Metropolitana, Mexico City 04530, Mexico; saracalderon_96@hotmail.com; 6Laboratory of Experimental Oncology, Instituto Nacional de Pediatria (INP), Mexico City 04530, Mexico; arochoa67@hotmail.com; 7Laboratory of Toxicology Genetics, Instituto Nacional de Pediatria (INP), Mexico City 04530, Mexico; rsantesp@pediatria.gob.mx (R.S.P.); vdoradog@hotmail.com (V.M.D.G.)

**Keywords:** antioxidant, artificial intelligent, androgen antagonists, 5α-reductase inhibitors, neuroprotection, myelin

## Abstract

Background: Two natural steroids derived from cholesterol pathways are testosterone and progesterone, androgen and antiandrogen receptor binding. Steroid androgen antagonists can be prescribed to treat an array of diseases and disorders such as gender dysphoria. In men, androgen antagonists are frequently used to treat prostate cancer and hyperplasia. Sex hormones regulate the expression of the viral receptors in COVID-19 progression, and these hormones may act as a metabolic signal-mediating response to changes in glucose and Reactive Oxygen Species (ROS). The objective of the present study is to use artificial intelligence (AI) applications in healthcare to predict the targets and to assess biological assays of novel steroid derivatives prepared in house from the commercially available 16-dehydropregnenolone acetate (DPA^®^) aimed at achieving the metabolic stability of glucose and steroid brain homeostasis. This suggests the introduction of aromatic or aliphatic structures in the steroid B-ring and D-ring. This is important since the roles of 5α-reductase and ROS in brain control of glucose and novel steroids homeostasis remain unclear. Methods: A tool prediction was used as a tuned algorithm, with the novel steroid derivatives data in web interface to carry out their pharmacological evaluation. The new steroidal derivatives were determined with neuroprotection effect using the select biomarkers of oxidative stress on induced hypoglycemic male rat brain and liver. The enzyme kinetics was established by the inhibition of the 5α-reductase enzyme on the brain myelin. Results: We used novel chemical structures to order the information of a Swiss data bank that allow target predictions. Biological assays suggest that steroid derivatives with an electrophilic center can interact more efficiently with the 5α-reductase enzyme, and by this way, induce neuroprotection in hypoglycemia model. All compounds were synthesized with a yield of 30–80% and evaluated with tool target prediction to understand the molecular mechanisms underlying a given phenotype or bioactivity and to rationalize possible favorable or unfavorable side effects, as well as to predict off-targets of known molecules and to clear the way for drug repurposing. Apart, they turned out to be good inhibitors for the 5α-reductase enzyme. Conclusions: The probed efficacy of these novel steroids with respect to spironolactone control appears to be a promising compound for future hormonal therapy with neuroprotection activity in glucose disorder status. However, further research with clinically meaningful endpoints is needed to optimize the use of androgen antagonists in these hormonal therapies in COVID-19 progression.

## 1. Introduction

Androgen antagonists are chemical substances that obstruct androgenic actions at the level of their binding sites. From a pharmacological point of view, 5α-reductase inhibitors (5ARIs) are not true antiandrogens, since their mechanism of action leaves circulating testosterone free to bind to androgen receptors. Androgens are implicated as having a critical role in the origin and progression of benign prostatic hyperplasia (BPH) and prostate cancer. Testosterone (structure 1) and progesterone (structure 2) are cholesterol-derived steroids, acting, respectively, as natural androgen and antiandrogen [1].



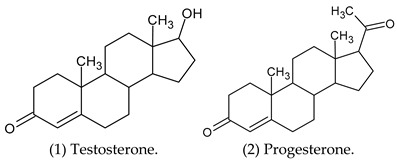



Antiandrogen agents are employed for the treatment of a range of sundry diseases reliant on androgen activity. Antiandrogens or androgen receptor blockers or testosterone blockers exert their effects by competitively inhibiting androgens from binding to their target receptors, or by suppressing the body’s overall production of these hormones [2]. These androgen antagonists are indicated for management of a range of androgen-dependent conditions, including gender dysphoria. They are a class of medications primarily used for the treatment of prostate cancer and BPH in men [3]. The combined management of advanced prostate cancer with androgen antagonists and castration has been reported to demonstrate better long-term survival and disease control than using castration alone [4]. The systemic administration of an androgen antagonist blocks androgen effects in both the intended target tissue and unintended off-target tissues; thus, it will antagonize the negative feedback loop through which androgens control their secretion via the hypothalamo-pituitary-testis (HPT) axis or their action through protein factors [5]. The impact of internally produced sex hormones on blood sugar regulation varies, and these variations are linked to an elevated risk of developing type 2 diabetes [6]. Nonetheless, ROS are capable of acting as cellular second messengers that modulate responses to changes in glucose and hormones [7]. The differential regulation of the viral receptors ACE2 and TMPRSS2 by sex steroids influences host susceptibility to viral entry, thereby influencing the extent of viral infection and subsequent clinical variability. Sex hormones may influence the risk of severe COVID-19 and mortality, potentially explaining the observed sex-based disparities in disease progression [8]. Hypoandrogenemia may represent a key predisposing factor for neurodegenerative conditions [9]. Multiple neurotransmitters contribute to the pathophysiology of these conditions, as dopamine and serotonin are key mediators within the neural reward circuits [10].

The significant tissue concentrations and important physiological functions of sulfonated steroids suggest their active involvement in metabolism. The authors hypothesize that a specific steroidogenic pathway exists for these compounds, initiated by the CYP11A1 enzyme’s conversion of cholesterol sulfate (CS) into pregnenolone sulfate (PregS), which subsequently undergoes 17α-hydroxylation to form 17OH-PregS [11]. Its highly water-soluble and generally hormonally inactive, serving primarily as circulating reservoirs or having distinct functions, for example, as neurosteroids.

Environmental androgen antagonists represent a growing topic of concern. Plant-derived compounds, exemplified by dehydropregnenolone acetate (structure 4), have been shown to exert antiandrogenic effects by antagonizing androgen receptors, suppressing de novo androgen synthesis, or a combination of both mechanisms [12].



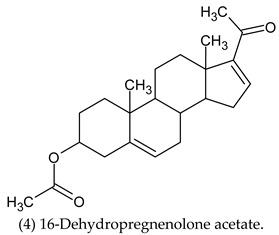



AR belongs to the nuclear receptor class (NR3C4), acting as a transcription factor whose activity is contingent on ligand binding [13]. AR antagonists represent the most common form of antiandrogen therapy; these compounds function by competitively blocking androgen-binding sites on target cells, thereby preventing the biological effects of androgens. Such antiandrogens are indicated for the treatment of various disorders, such as prostate cancer, BPH, hirsutism, acne, and male-pattern baldness. Nonetheless, the intolerable physicochemical characteristics and absorption, distribution, metabolism, and excretion (ADME) properties of AR-binding molecules were less applicable in clinical areas [14]. This has allowed the discovery of more highly effective drug candidates.

The antiandrogen-mediated suppression of androgen synthesis follows a distinct pathway for every agent. For instance, the drug acts as a dual-target inhibitor, antagonizing the androgen receptor and suppressing androgen synthesis through the inhibition of key steroidogenic enzymes, specifically cytochrome P450 and 17,20-lyase, which mediates the synthesis and breakdown of steroids, such as testosterone precursors. Consequently, overall testosterone secretion from the testes and adrenal cortex is attenuated [15]. Gonadotrophins, the key regulators of androgen production secreted by the pituitary, are susceptible to the effects of antiandrogens.

The mechanism by which antiandrogens inhibit gonadotropin release involves the down-regulation of GnRHRs within the pituitary gland. A reduction in the number of GnRHRs attenuates the binding affinity of gonadotropin-releasing hormone (GnRH). This hormone stimulates the secretion of luteinizing hormone (LH) and follicle-stimulating hormone (FSH), which are gonadotropin hormones. LH binds to receptors on Leydig cells of the testes and theca cells of the ovaries, triggering the secretion of testosterone. Consequently, the absence of GnRH binding prevents the induction of testosterone synthesis within the gonads.

A subset of prostate cancer cells exhibits androgen-dependent proliferation. Androgen deprivation therapy (ADT) involves the use of antiandrogens which act as competitive inhibitors for androgen receptors, effectively blocking the androgen-signaling axis critical for prostate cancer cell survival and proliferation [16]. Systemic antiandrogens exhibit off-target effects by blocking androgen receptors in various tissues, thereby disrupting the negative feedback mechanisms of the HPT axis and altering systemic androgen regulation. While antiandrogen monotherapy demonstrates a better quality-of-life and side-effect profile compared to combined approaches, its efficacy in complete androgen deprivation may be diminished.

Monotherapy is often preferred by men as it is less likely than combined therapies to diminish libido [17]. An effect of novel coronavirus disease 2019 (COVID-19) on male sex hormones is the decrease in testosterone by 3.94 nMol/L, a figure obtained through comparison with healthy [18] men, and this invariably affects male fertility. Nevertheless, the demise of more men than women during the SARS-CoV-2 pandemic later demonstrated the protective effects of estrogen may play a role in this aspect [19]. Prostate cancer cells treated with antiandrogenic drugs can develop castration-resistant prostate cancer (CRPC), a condition where tumor cells acquire the mechanisms for survival and proliferation independent of androgen receptor (AR) signaling, while androgen-dependent cells undergo apoptosis. However, a subset of cancer cells adapts to survive in low-androgen conditions, and these are the cause of cancer relapses subsequent to the initial resolution of clinically detectable prostate cancer for a period of several years [20]. Resistance to therapy involves a variety of mechanisms, notably AR amplification and overexpression, androgen receptor gene mutations, the presence of constitutively active AR variants, intratumoral steroidogenesis as well as AR activation by non-androgen ligands including specific AR mutations (L702H, W742L/C, H875Y, F877L, and T878A/S), which are often found in individuals with treatment-resistant disease [21]. Nonetheless, naturally occurring compounds possess anti-androgenic activity capable of attenuating the DHT-driven secretion of PSA and their intracellular dispersal enables the determination of each phytochemical’s capacity for cellular internalization and nuclear accumulation, where the nucleus serves as the site of the proposed transcriptional modulation, subsequent to phytochemical-sex steroid receptor binding [22]. However, studies have shown that the unnatural enantiomers of 17-keto androgen neurosteroids exhibit greater therapeutic potential, with superior in vivo potency and a longer duration of action than their natural counterparts [23]. Indeed, gene determination is carried out in the neurons and glial cells of the brain using quantitative real-time PCR as well as in receptor-ligand interaction, axon pathfinding, cytokine signaling pathway and oncogenic metabolic reprogramming. Genes involved in neuroreceptor ligand interactions, such as those encoding GABA_A_ and GABA_B_ receptors and nuclear core proteins, were conspicuously absent from the list of top upstream regulators [24].

5α-Reductase inhibitors (5-ARIs) are a class of antiandrogenic agents that prevent the enzymatic conversion of testosterone to the more potent androgen, dihydrotestosterone (DHT) [25], by targeting the 5α-reductase enzyme (Figure 1).

Dihydrotestosterone (DHT) exhibits approximately three- to five-fold greater androgenic potency compared to testosterone. The distinguishing feature of antiandrogenic agents is their specific inhibition of DHT-mediated pathways, without impacting the activity of alternative androgens. DHT is the primary androgen responsible for the formation of the male external genitalia during fetal development and subsequent prostate development. Androgen deprivation via castration significantly decreased the density of apical dendritic spines in mice, which was accompanied by more pronounced ultrastructural defects in excitatory synapses [26] within the tissue.

Finasteride 7, a 5α-reductase inhibitor, is commonly prescribed for the management of BPH [27] and androgenetic alopecia (AGA) [28], due to its mechanism of action involving the suppression of dihydrotestosterone (DHT) production, which consequently attenuates prostatic cell proliferation. The genesis of benign prostate hyperplasia (BPH) depends on two factors: testicular androgen and the aging process. The most important androgen in the prostate is DHT [29]. The flutamide blocks the action of both endogenous and exogenous testosterone and can block androgen binding and subsequent nuclear translocation in prostatic tissue [30].



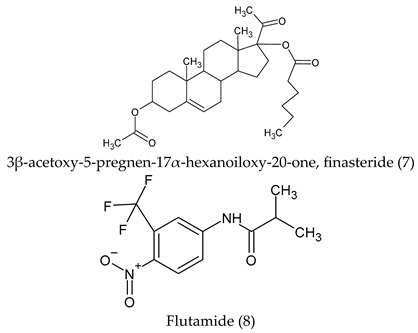



There are multiple 5α-reductase isozymes that exhibit distinct expression profiles in various tissues and cell types, such as epithelial and stromal compartments [31]. Young rats exhibit a significantly greater capacity for oligodendrocyte-mediated 5α-reduction in testosterone compared to adult rats [32]. Consequently, a more potent dual 5α-reductase inhibitor would lead to a greater reduction in both circulating and intraprostatic DHT levels and would be effective in managing BPH and other dihydrotestosterone-related disorders, such as gender incongruence. This incongruence can lead to an important clinical distress culminating in craving for gender affirmation. Sex reassignment, also known as transition, is a process of changing the sex of a person by surgical procedures, hormonal therapy and others to achieve a desired gender aspiration. Standard-of-care protocols for transgender women’s medical transition involve the use of hormonal therapy and, on a case-by-case basis, surgical interventions [33]. Antiandrogens exert their effects through diverse pathways to attenuate androgenic activity. However, insufficient evidence exists to determine the most effective antiandrogen for feminization outcome [34]. In any case, spironolactone (30) and cyproterone acetate (11) are frequently administered in the context of gender-affirming hormone therapy (GAHT) for transfeminine individuals, with the objective of suppressing endogenous testosterone concentrations to a target female reference range [35,36]. However, the administration of supplementary hormones enhances breast cancer susceptibility by stimulating ductal and lobular development and augmenting fat deposition within the mammary tissue [37]. Testosterone administration is associated with the highest risk of metabolic syndrome in transmasculine individuals, whereas estradiol treatment in transfeminine individuals confers the lowest risk [38].



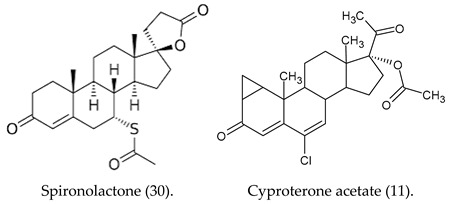



Conversely, flutamide and nilutamide, which are peptide-based inhibitors, disrupt AR–protein interactions by binding to the receptor surface. This mechanism-based strategy offers superior receptor blockade potential relative to conventional ligand-receptor binding methods. Indeed, widespread progesterone receptors (PRs) expression was detected in all the brain regions following transient middle cerebral artery occlusion (tMCAO). Levels of the endogenous PR ligands, progesterone and 5α-dihydroprogesterone were found to increase rapidly six hours post-ischemia, making the authors propose that the surge is part of the brain’s intrinsic neuroprotective response [39]. Key steroidogenic enzymes, analogous to those expressed in peripheral endocrine glands, mediate the central nervous system’s capacity for neurosteroidogenesis. The brain contains significant concentrations of dehydroepiandrosterone (DHEA) (31), which functions as a major locally synthesized neurosteroid. DHEA is synthesized via a metabolic route driven by oxidative stress, a pathway that operates independently of the peripheral cytochrome P450 17α-hydroxylase/17,20-lyase (CYP17A1) catalytic activity [40]. The mitochondrial enzyme, 17β-Hydroxysteroid dehydrogenase (17β-HSD10), which exists as a homotetramer with ability to oxidize neuroactive steroids, is capable of binding to amyloid-β and is implicated in neurodegeneration [41].



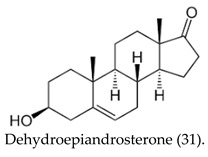



In fact, endocrine-disrupting chemicals (EDCs) present in the environment can mimic, block, or otherwise disrupt the body’s natural hormones. These substances are classified as endocrine disruptors (EDs) and disrupt the function of 11β-hydroxysteroid dehydrogenase (11βHSD), which regulates the intracellular conversion of inactive glucocorticoids (e.g., cortisone) to their active forms (e.g., cortisol). The enzyme exhibits two distinct isoforms and is widely expressed across multiple tissues. The potential of 11βHSD type 1 inhibitors as novel pharmacological interventions for metabolic syndrome and comorbid depressive disorders is being assessed [42]. This highlights the necessity for next-generation antiandrogens engineered for increased AR affinity compared to natural androgens, ensuring pure antagonistic activity and enabling more effective single-agent androgen blockade. Indeed, research suggests that structural alterations to the steroid B-ring and the C-17 side chain are critical determinants of efficacy in hormonal therapy [43], and considerable attention has been devoted to identifying and optimizing novel antiandrogen compounds, which could manage or mitigate the aforementioned clinical disorders, achieved through the strategic incorporation of aromatic or aliphatic moieties into the steroid B- and D-rings [44]. The novel steroid synthetics (structure 32) were obtained with structural characterization and modifications from 16-Dehydropregnenolone Acetate^®^ (DPA). The homologous compounds were analyzed and characterized with NMR ^1^H and ^13^C, UV, IR spectroscopy, mass spectrometry, melting point, and thin layer chromatography. The steroid derivatives’ purity was ± 1 degree and yields were between 30% and 80%, depending on the novel chemical compound and functional group inserted.

General route of chemical synthesis:



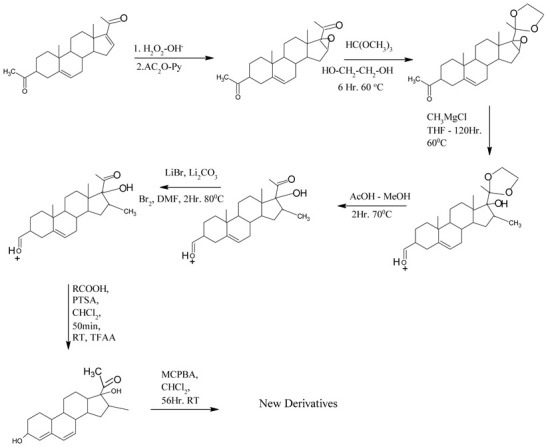



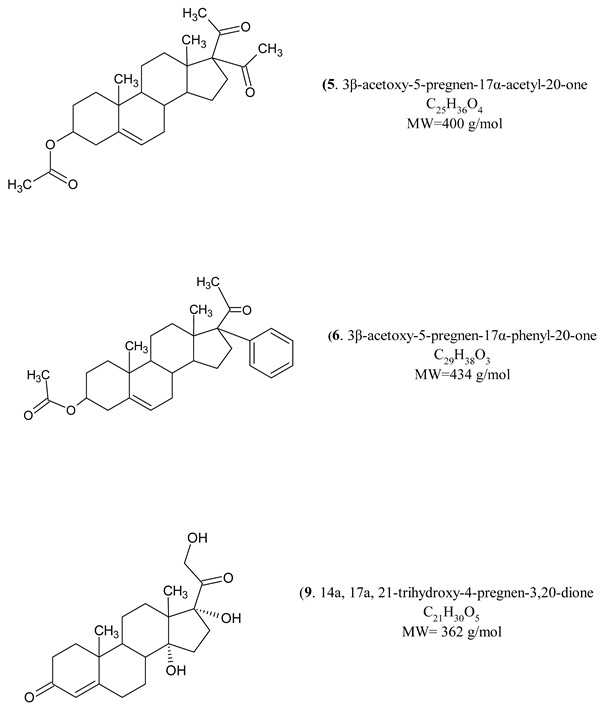



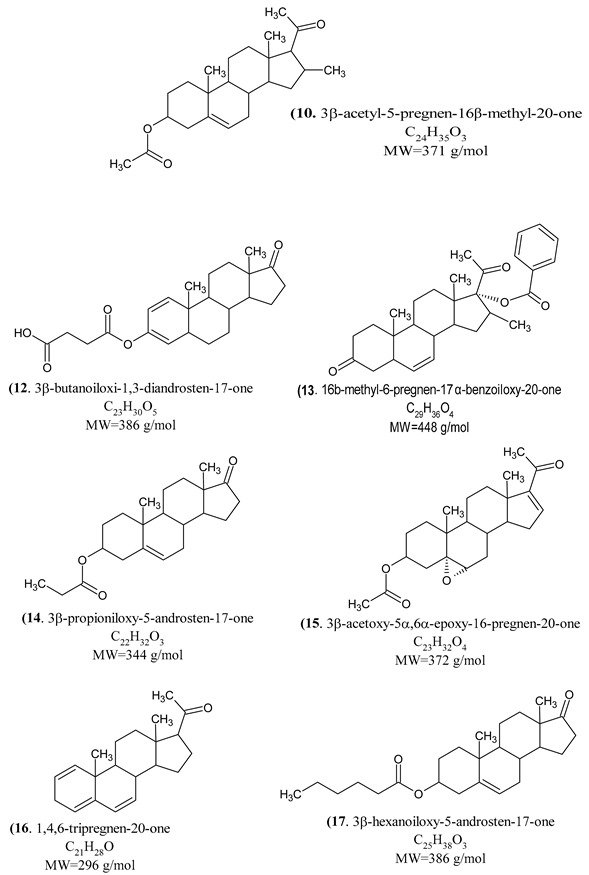



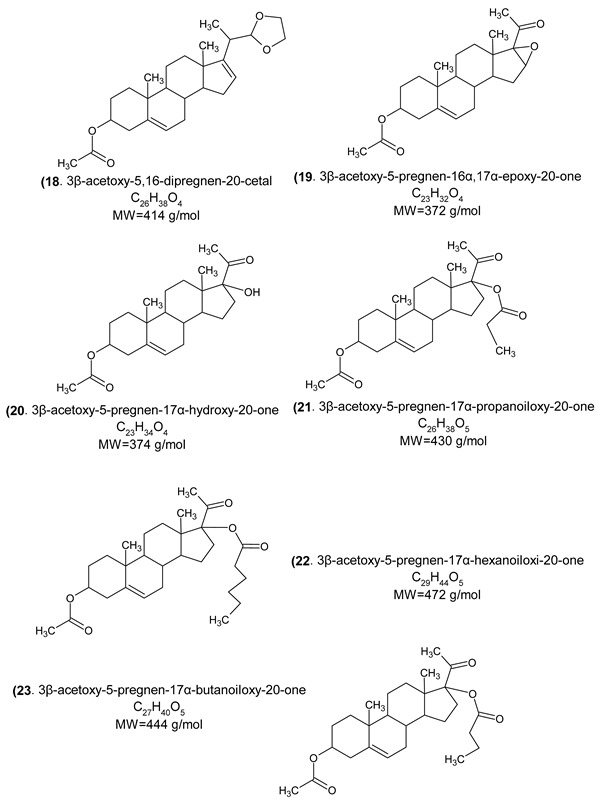



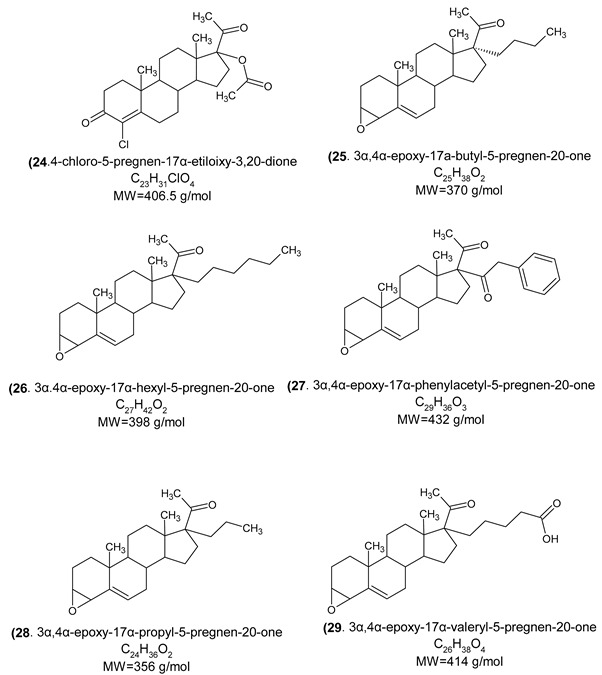



An AI-based target prediction model was leveraged to inform the design of novel steroid molecular structures, using biomarker data accessed via the Swiss biobank system. We utilized a fine-tuned algorithm and novel data, accessible via a web interface, to apply artificial intelligence in healthcare to predicting human steroid therapy targets (Figure 2). To predict combination therapies and probable side effects, Table 1 was utilized to present the Swiss Target Prediction data for natural and novel synthetic steroids. The utility of this tool lies in its ability to clarify the molecular mechanisms associated with a given phenotype or bioactivity, aid in the rationalization of possible side effects, the prediction of off-target interactions, and the advancement of drug repurposing initiatives. A ligand-based approach was employed for predictions, leveraging the similarity between the query molecule and known ligands associated with numerous protein targets in human metabolic pathways.

The cyclopentane-fused hydrogenated phenanthrene chemical structure is what confers the antiandrogenic steroid drugs their antagonistic property. Potent androgen receptor inhibition in conjunction with curative radiotherapy may improve outcomes for men diagnosed with high-risk localized prostate cancer [45]. Conversely, recent data indicate that steroids confer significant survival benefits among hospitalized COVID-19 patients.

On the contrary, the therapeutic benefit of dexamethasone in severe COVID-19 is mediated, in part, by dampening the inflammatory neutrophil response; this involves down-regulating type I IFN signaling, increasing IL-1R2 expression, and fostering an expansion of immature, immunosuppressive neutrophil states from information receivers into information providers. Male patients exhibited a higher baseline proportion of IFN-active neutrophils and a more pronounced steroid-induced expansion of immature neutrophils, factors that potentially influence clinical outcomes [46] and recovery trajectories.



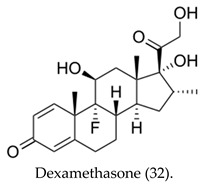



The drug’s benefits are potentially mediated by non-canonical androgen receptor activity, an expected pharmacological response in the treatment of adult clinical disorders using antiandrogens. The modulation of androgen signaling presents a promising strategy for viral blockade; consequently, antiandrogens could be employed as a prophylactic intervention in early-stage hospitalized COVID-19 patients [47] and, in general, healthy population. A quantitative meta-analysis of 21 relevant studies involving 9922 patients (6265 males, 3657 females) yielded a pooled estimate (e.g., odds ratio) of 0.52 (95% CI [0.34, 0.80]), demonstrating a significant reduction in mortality among COVID-19 patients receiving steroids compared to the non-steroidal group [48]. Although significant clinical equipoise persists regarding steroid administration in COVID-19 patients, current data demonstrate that their use effectively lowers the mortality risk in hospitalized individuals. However, continued investigation through additional clinical evidence is necessary.



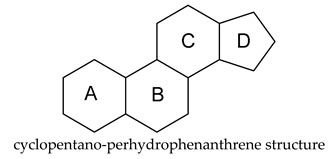



Broadly speaking, when the properties of endogenous steroids and the novel synthetic steroids were determined, it was found auspicious to incorporate aliphatic group with carbonyl in ring A, chloride or epoxide in ring B and ester-aliphatic group in ring D. Aromatic substitutions at the D-ring induce non-androgen receptor-mediated effects, specifically activating thromboxane and vascular pathways, and upregulating inducible nitric oxide synthase expression. Nonetheless, current steroidal antiandrogens face limitations in clinical utility across applications like gender-affirming hormone therapy (GAHT) and prostate cancer treatment due to their suboptimal physicochemical and pharmacokinetic profiles. A lack of robust evidence currently impedes the determination of optimal efficacy and safety in these populations. Hence, there is a critical need for additional research with clinically meaningful endpoints to optimize antiandrogen utilization in these contexts and investigate their potential role in mitigating severe COVID-19 outcomes.

The COVID-19 pandemic is estimated to have caused over 6.5 million deaths globally since its onset. Clinical observations indicate that male sex is associated with increased infection severity and higher mortality rates, a disparity potentially linked to the effects of androgens. A notable gender disparity exists in COVID-19 outcomes, as men exhibit more severe infections and higher mortality rates. Studies have indicated that androgen deprivation may be protective, potentially lowering the incidence and severity of COVID-19 infection comorbidity in susceptible patients [49].

Biological Evaluation: Toxicity is initiated by the augmented production of reactive oxygen species (ROS); when ROS generation surpasses the capacity of the endogenous antioxidant defense systems, oxidative stress ensues. Reactive oxygen species (ROS) and reactive nitrogen species (RNS) are the main quantifiable markers of oxidative stress in physiological and pathological states, potentially aiding in the assessment of inflammatory disorders and overall physiological well-being. Selected biological markers, including glutathione (GSH), lipid peroxidation (LPO), 5-hydroxyindoleacetic acid (5-HIAA), dopamine, and catalase activity, are commonly employed as oxidative stress indicators to evaluate the balance between pro-oxidant and antioxidant status. While efforts are underway to synthesize novel antiandrogens by functionalizing the steroid B- and D-rings with aromatic or aliphatic structures, key mechanistic questions persist regarding the involvement of 5α-reductase and ROS in brain glucose regulation and the metabolism of these new compounds.

## 2. Material and Methods

Insulin^®^, DPA^®^, DMSO, Serotonin metabolite, Dopamine, GSH, Tiobarbituric acid, Octyl suphate, EDTA, O-Phthaldehyde, NADPH, MeOH, and ATP were acquired in Sigma-Aldrich (St. Louis, MO, USA). NaCl, KH_2_PO_4_, Citric acid, Ascorbic acid, TCA, HClO_4_, H_2_SO_4_, and HCl were acquired in Merck Darmstadt (Darmstadt, Germany).

### 2.1. Experimental Animals

Experimental animals were sourced from the certified laboratory animal breeding facility at CINVESTAV NORTE, Mexico City, Mexico. Rats were group-housed (n = 5 per cage) in meshed plastic enclosures under a 12 h light/12 h dark cycle (12:12 L:D), maintained under natural environmental conditions. Animals were provided Let as italics access to a standard maintenance diet of granulated rodent feed (Purine 5001^®^ Lab Rodent Diet 5001, St. Louis, MO, USA) and fresh drinking water throughout the experimental period. To minimize stress and allow for the adaptation to the facility’s environment, animals were housed for a 14-day acclimatization period with ad libitum access to feed and water before the experiment commenced. The animal study protocols adhered to the principles outlined in the National Institutes of Health (NIH) Guide for the Care and Use of Laboratory Animals (Publication No. 8023, revised 1978). The protocol was reviewed and approved by The Ethics and Investigation Committee of Laboratory Animals Care of the National Institute of Pediatrics (Code: 026/2024) on 19 June 2024.

### 2.2. Investigational Products

A total of twenty male Wistar rats, weighing approximately 180 g each at the start of the experiment, were utilized and randomly assigned to four experimental groups (n = 5 per group). Animals in the experimental groups received a single dose of either insulin (50 U/kg body weight BW) or a randomly selected steroid (11 mg/kgBW). Treatments were administered via intraperitoneal (i.p.) injection. Animals were assigned to one of four groups: Group 1 received no treatment (basal control); Group 2 (vehicle control) received dimethyl sulfoxide (DMSO) and insulin; Group 3 received a single dose of insulin combined with steroid 12; and Group 4 received four doses of insulin combined with steroid 12, administered every 36 h (Table 2).

### 2.3. Experimental Design

At the conclusion of the treatment period, animals were euthanized 60 min post-insulin administration. To avoid the confounding effects of anesthetics on oxidative stress markers, animals were humanely euthanized by decapitation, and blood was collected immediately thereafter. To assess relevant biomarkers, blood was analyzed for triglycerides and glucose. Following rapid tissue extraction post-euthanasia, brain and liver samples were processed to measure lipid peroxidation (TBARS), 5HIAA, Dopamine (DA), glutathione (GSH) and catalase. Following extraction, the brain was microdissected into its constituent parts: hemispheres, cerebral cortex, and medulla oblongata. The dissected brain areas were kept frozen at −20 °C in physiological saline (0.9% NaCl) until the time of assay. The homogenization of the hemispheres, cortex and medulla/oblongata as well as the liver were performed using 3 mL of phosphate buffer solution 0.05 M pH 7.5. The homogenates were utilized for the ROS biomarkers with TBARS, 5-HIAA, dopamine, GSH, and catalase determination. The assay of these biomarkers was based on methods that were formerly validated. The preservation of the homogenized tissues was achieved by keeping them at −20 °C until they were analyzed. Animal handling and experimental protocols were performed in compliance with the relevant national and international ethical guidelines and regulations. This study protocol received the approval of the committee for the management of laboratory animals, reference number 022/26.

### 2.4. GSH Assessment

The determination of GSH levels followed the established technique developed by Hissin and Hilf [50]. A reaction mixture was prepared in a test tube by combining 20 µL of the supernatant with 1.8 mL of phosphate buffer (pH 8.0, 0.2% EDTA) and 100 µL of 1 mg/mL ortho-phthaldehyde in methanol. Incubation of the mixture proceeded at room temperature for 15 min, protected from light. A Perkin Elmer LS 55 spectrophotometer (Beaconsfield, UK) was employed to determine the level of GSH, utilizing excitation and emission wavelengths of 350 nm and 420 nm, respectively. GSH concentrations were calculated by inference from a previously established standard curve using FL Win Lab software v.4.00.02, with results reported in µM per gram of tissue.

### 2.5. 5-HIAA Assay and Assessment

The quantification of 5HIAA utilized the supernatant from brain tissue homogenized in HClO_4_ and centrifuged (9000 rpm for 10 min; Hettich Zentrifugen Mikro 12–42, Tuttlingen, Germany). This procedure was a modification of the technique described by Beck et al. [51]. A specific volume of the supernatant was transferred into a test tube containing 1.9 mL of 0.1 M acetate buffer (pH 5.5). Each sample was subsequently analyzed using a Perkin Elmer LS 55 fluorometer (Beaconsfield, UK) at an excitation wavelength of 296 nm and an emission wavelength of 333 nm. The assessment utilized FL Win Lab software, version 4.00.02. The 5-HIAA values were determined by interpolation from a previously established standard curve and expressed as µM per gram of wet tissue.

### 2.6. Dopamine (DA) Assay and Assessment

We quantified DA in the resulting supernatant after homogenizing tissue in HClO_4_ and centrifuging the mixture at 5000 rpm for 10 min (Hettich Zentrifugen Mikro 12–42). The assay was based on a modified protocol reported by Guyon et al. [52]. An aliquot of the HClO_4_ supernatant, and 1.9 mL of buffer (0.003 M octyl-sulfate, 0.035 M KH_2_PO_4_, 0.03 M citric acid, 0.001 M ascorbic acid), were placed in a test tube. The mixture was incubated for 5 min at room temperature in total darkness, and subsequently, the samples were read in a spectrofluorometer (Perkin Elmer LS 55, Beaconsfield, UK) with 282 nm excitation and 315 nm emission lengths. Results were processed using FL Win Lab software (v. 4.00.02). Concentrations were inferred from a standardized curve and reported in nmol/g of wet tissue.

### 2.7. Assessment of Lipid Peroxidation

TBARS levels were determined using a modified method derived from the techniques of both Gutteridge and Halliwell [53] and Guzman et al. [54]. Homogenates (1 mL) of the hemispheres, cortex, medulla/oblongata, and liver tissues prepared in 0.05 M PBS (pH 7.4) were mixed with 2 mL of TBA solution. The TBA reagent solution was prepared by dissolving 1.25 g of TBA and 40 g of trichloroacetic acid (TCA) in a mixture of 6.25 mL of concentrated HCl and 250 mL of deionized water. The solution was subjected to a 30 min heating period at boiling point using a Thermomix 1420 (Anaheim, CA, USA). Following heating, the solution was immersed in an ice bath for 5 min to facilitate cooling and then centrifuged at 700× *g* for 15 min using a Sorvall RC-5B Dupont centrifuge (Kansas City, MO, USA). The absorbance of the brain tissue supernatant was measured in triplicate at 532 nm on a Beckman DU 340 spectro-photometer (Fullerton, CA, USA) to quantify TBARS concentration. The results of the TBARS assay were quantified as µM of malondialdehyde (MDA) per gram wet tissue.

### 2.8. Measurement of Catalase (CAT)

Catalase (CAT) activity in the hemispheres, cortex, and medulla/oblongata was measured using a Cayman Chemical Catalase Assay Kit (Ann Arbor, MI, USA) following the modification outlined in Hadwan et al. [55]. Results are reported as nmoles/g tissue/min.

### 2.9. Cerebral Cortex Homogenates

Cerebral cortices were collected from decapitated male Wistar rats. The tissues were homogenized in 0.25 M Tris-HCl (pH 7.4) and cryopreserved prior to analysis.

### 2.10. Purification of Myelin

The procedure for the purification of myelin followed that of Melcangi et al. [32], albeit with minor modifications. The final pellet was harvested and resuspended in 0.1 M phosphate-buffered saline (PBS), pH 7.5.

### 2.11. Kinetic Enzymatic of 5α-Reductase Assay

The incubation was performed in PBS 0.1 M pH 7.5 in presence of NADPH system 4 mM, Testosterone 4 mM, protein myelin 1 mg, novel steroid 0.5, 1.0, 2.0 and 4.0 mM to time incubation of 30 min to 37 °C [56]. To stop the reaction was used trichloroacetic acid 10% and were read to 340 nm in Molecular devices spectro maxplus (San Jose, CA, USA) and Software SoftMaxPro 6.0. Apparent Km and Vmax values were estimated by fitting the reaction rate data to the Michaelis-Menten equation using non-linear regression (or a Lineweaver-Burk plot, Eadie-Hofstee plot, etc.) (Table 3). All kinetic parameters were determined from quadruplicate assays.

### 2.12. Statistical Analysis

The data were analyzed using descriptive statistics and presented in tables and graphs to show central tendency and spread. Statistical analysis of the biochemical indicators involved comparing the control group with the experimental groups. Parametric assumptions were assessed for variance homogeneity; accordingly, group differences were determined using Analysis of variance (ANOVA) or the non-parametric Kruskal–Wallis test. When a significant difference was detected, Tukey–Kramer or Steel-Dwass post hoc tests were applied for pairwise comparisons. Any associated probability value α < 0.05 was regarded statistically significant. SAS Systems JMP v12 software [57], was used in performing the analysis.

## 3. Results

All test compounds exhibited significant 5α-reductase inhibitory activity compared to the spironolactone control group (*p* < 0.05) (Figure 3, Figure 4, Figure 5, Figure 6 and Figure 7). The observed efficacy of these novel steroids, relative to the spironolactone control group, indicates significant potential as a promising avenue for future hormonal therapeutic interventions in clinical practice (Table 4).

## 4. Discussion and Conclusions

In rats, the 5α-reductase (5α-R) enzyme is found in multiple Central Nervous System (CNS) structures, with the highest activity concentrated in the myelin sheaths. Poletti and colleagues reported that while male rats exhibit similar apparent Michaelis-Menten constant (Km) values, the maximum reaction velocity (Vmax) values derived from myelin were consistently distinct from other CNS structures [56]. However, this study showed opposite results with Km and Vmax, (Table 3). The assessed novel steroids would be effective in managing disorders linked to dihydrotestosterone (DHT).

Future antiandrogenic steroid drugs are likely to be antagonists, featuring the cyclopentano-perhydrophenanthrene skeleton. Androgen receptor (AR) antagonists exhibit an alternative mechanism of action, which represents a key mechanism responsible for favorable clinical outcomes in adult populations. The structure-activity relationship (SAR) analysis of novel synthetic steroids indicates that the incorporation of aromatic or ester-aliphatic groups in ring D, chloride in ring B, and aliphatic groups with a carbonyl moiety in ring A is advantageous. The incorporation of aromatic moieties into the D-ring potentiated the activity of 5α-reductase. Nonetheless, future studies with robust, clinically meaningful endpoints are necessary to fully optimize antiandrogen use and determine the risk factors, including those related to COVID-19, in hormonal therapies for transgender women and prostate cancer patients.

Indeed, the initial step of the SARS-CoV-2 infection involves the ability of the viral spike (S) protein to recognize and attach to the host cell’s angiotensin-converting enzyme 2 (ACE2) transmembrane protein; this is a prerequisite for subsequent S protein cleavage and activation by the cell-surface-associated transmembrane protease serine 2 (TMPRSS2). Elevated TMPRSS2 expression driven by androgen signaling in prostate cancer cells supports the hypothesis that this mechanism may increase men’s risk of severe COVID-19 outcomes via increased lung TMPRSS2 levels [58]. In any case, the therapeutic potential and safety profile of antiandrogens in the context of novel coronavirus infection necessitate additional research and clinical trials. In this biological work, we found hypoglycemia to be high due to a significant decrease in glucose levels in insulin-treated experimental animals and the use of steroids in the treatment of the young animals with induced hypoglycemia increased glutathione (GSH) in cortex, hemispheres and medulla oblongata, and a partial decrease of 5-HIAA in the same tissues (Figure 3, Figure 4, Figure 5, Figure 6 and Figure 7). These additive effects indicate a reduction in oxidative stress due to the administration of steroids and insulin and suggest that antiandrogen pretreatment protected cells from oxidative stress. The present results coincide with previous work by Holmes et al. [59], which posits that the neuroprotective properties of androgens are contingent upon oxidative stress levels; while protective when stress is low, androgens appear to worsen oxidative damage when stress is high. Therefore, the development of novel steroid therapeutics capable of mitigating oxidative stress could prove beneficial in preventing the progression of androgen-dependent cancers [60]. Therefore, the observed antioxidant effects offer a scientific basis for designing clinical studies to reduce oxidative stress and potential CNS side effects of hormonal therapy in patients, as the optimal antiandrogen for balancing feminization and COVID-19 management remains undefined.

Owing to the study’s limitations, these findings require verification through large-scale studies that investigate the influence of COVID-19 risk factors on sex hormone levels and fertility in both males and females.

## Figures and Tables

**Figure 1 cimb-47-01059-f001:**
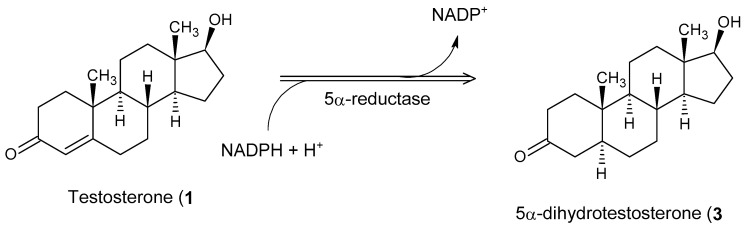
Antiandrogens 5α-reductase.

**Figure 2 cimb-47-01059-f002:**
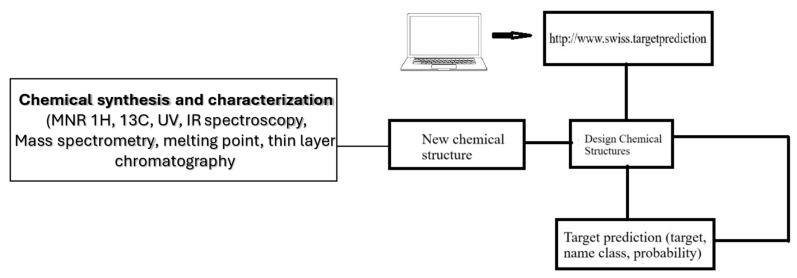
Predict human steroid therapy targets by https://www.swisstargetprediction.ch/ (accessed on 15 December 2015).

**Figure 3 cimb-47-01059-f003:**
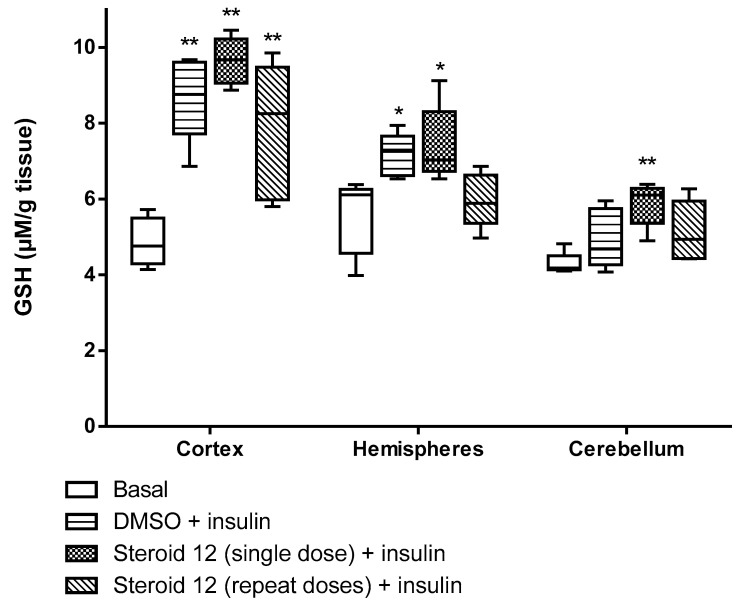
GSH levels in rat brain regions treated with novel steroid 12 and insulin. Cortex: DMSO + Insulin, Steroid 12 (single dose) + Insulin, Steroid 12 (repeated doses) + Insulin vs. Basal ** *p* < 0.001. Hemispheres: DMSO + Insulin, Steroid 12 (single dose) + Insulin vs. Basal * *p* < 0.05. Cerebellum: Steroid 12 (single dose) + Insulin vs. Basal ** *p* < 0.001.

**Figure 4 cimb-47-01059-f004:**
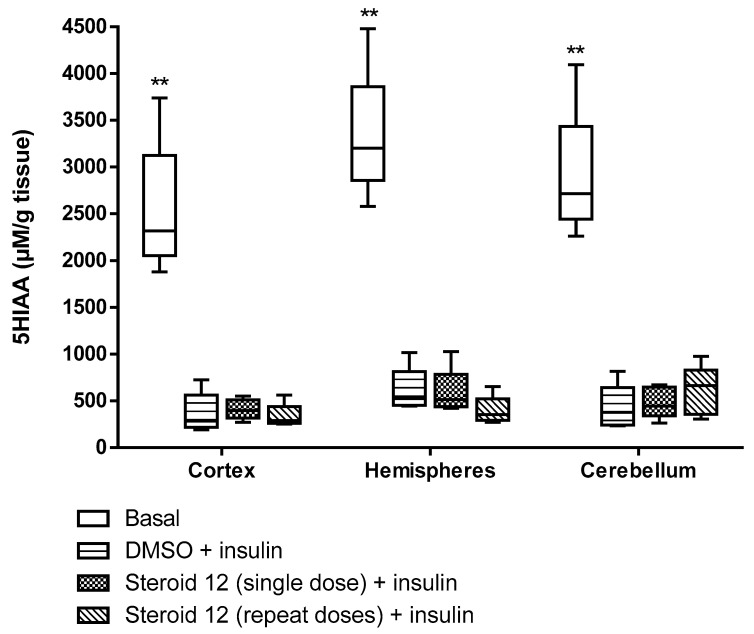
5HIAA levels in rat brain regions treated with novel steroid 12 and insulin. Cortex: Basal vs. DMSO + Insulin, Steroid 12 (single dose) + Insulin, Steroid 12 (repeated doses) + Insulin ** *p* < 0.001. Hemispheres: Basal vs. DMSO + Insulin, Steroid 12 (single dose) + Insulin, Steroid 12 (repeat doses) + Insulin ** *p* < 0.001. Cerebellum: Basal vs. DMSO + Insulin, Steroid 12 (single dose) + Insulin, Steroid 12 (repeat doses) + Insulin ** *p* < 0.001.

**Figure 5 cimb-47-01059-f005:**
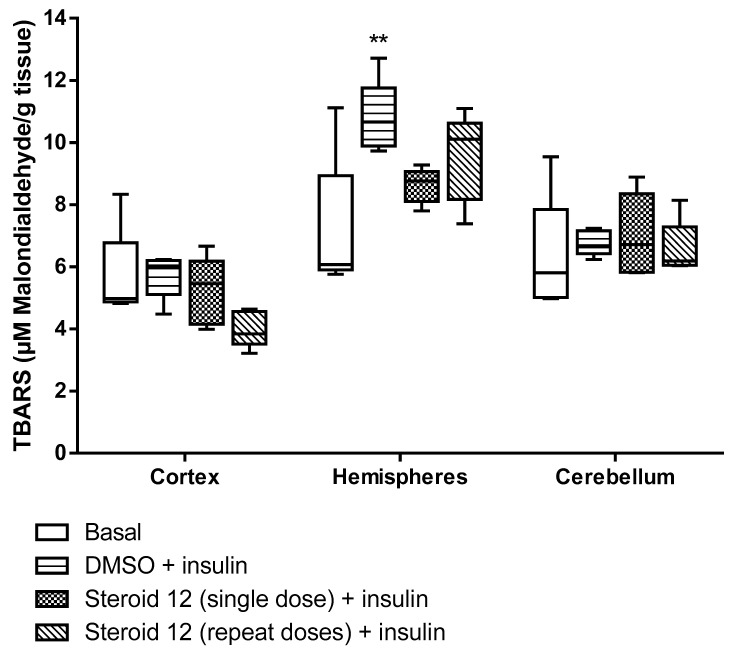
Lipoperoxidation levels in rat brain regions treated with novel steroid 12 and Insulin. Cortex and Cerebellum *p* = N.S. Hemispheres: DMSO + Insulin vs. Basal ** *p* < 0.001.

**Figure 6 cimb-47-01059-f006:**
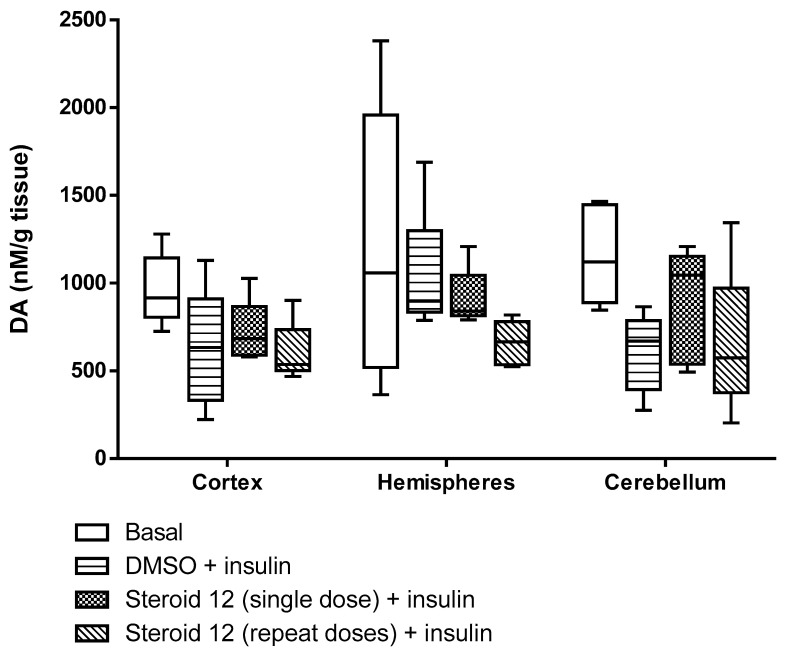
Dopamine levels in rat brain regions treated with novel steroid 12 and insulin. Cortex, Hemispheres and Cerebellum *p* = N.S.

**Figure 7 cimb-47-01059-f007:**
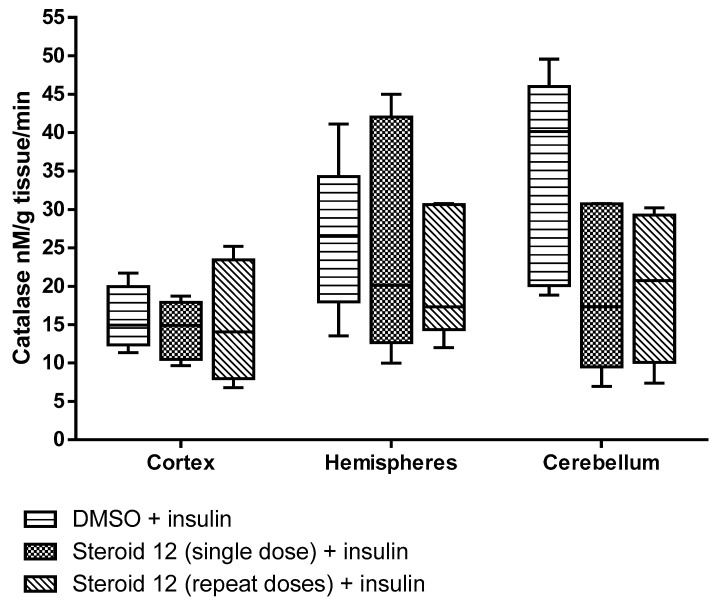
Catalase activity in rat brain regions treated with novel steroid 12 and insulin. Cortex, Hemispheres and Cerebellum *p* = N.S.

**Table 1 cimb-47-01059-t001:** Main target prediction on natural and novel synthetic steroids (**1**–**31**).

Steroid or Structure	Target Prediction
Testosterone (1)	Androgen receptor, Cytochrome P450 19A1, Glucocorticoid receptor, Corticosteroid-binding globulin, Sigma opioid receptor, Testis-specific androgen-binding protein.
Progesterone (2)	Steroid 5α-reductase 2, Cytochrome P450 2C9, Cytochrome P450 2C19.
5α-Dihydrotestosterone (3)	Androgen receptor, Cytochrome P450 19A1, Corticosteroid-binding globulin, Testis-specific androgen-binding protein, Estradiol 17-β-dehydrogenase 3, Glucose-6-phosphate 1-dehydrogenase.
16-Dehydropregnenolone Acetate (4)	Cytochrome P450 17A1.
(5)	Cytochrome P450 19A1, Cytochrome P450 17A1, 11-β-hydroxysteroid dehydrogenase-1, Protein tyrosine phosphatase 1B, Glycogen synthase kynase-3 beta, Androgen receptor.
(6)	Cytochrome P450 19A1, Cytochrome P450 17A1, 11-β-hydroxysteroid dehydrogenase-1, Protein tyrosine phosphatase 1B, Purinergic receptor P2Y1, Nicotinamide Phosphoribosyltransferase, Orexin receptor 2, Nitric oxide synthase inducible.
Finasteride (7)	Androgen receptor, Mineralocorticoid receptor, Glucocorticoid receptor, Progesterone receptor, Corticosteroid-binding globulin, Sigma opioid receptor, Testis-specific androgen-binding protein, Cytochrome P450 19A1.
Flutamide (8)	Androgen receptor, serotonin 6 (5-HT6) receptor.
(9)	Glucocorticoid receptor, Corticosteroid-binding globulin, Testis-specific androgen-binding protein, Androgen receptor, Mineralocorticoid receptor, progesterone receptor.
(10)	Cytochrome P450 19A1, Cytochrome P450 17A1.
Cyproterone acetate (11)	Androgen receptor, Glucocorticoid receptor, Adenoside A1 receptor, µu Opioid receptor, Cytochrome P450 2C19.
(12)	Thromboxane A2 receptor, Prostanoid DP receptor, Thromboxane A synthase, G protein-coupled receptor 44, Epoxide hydratase, steroid 5α-reductase 2, Prostanoid EP4 receptor
(13)	Protein-tyrosine phosphatase 1B, T-cell protein-tyrosine phosphatase, Cytochrome P450 19A1, Androgen receptor, steroid 5α-reductase 1, 3β-Hydroxysteroid dehydrogenase/delta 5-4 Isomerase type 1.
(14)	Cytochrome P450 19A1, Cytochrome P450 17A1, 11β-hydroxysteroid dehydrogenase 1, Steroid 5α-reductase 2, Androgen receptor, Progesteron receptor.
(15)	Androgen receptor, Cytochrome P450 19A1, Progesterone receptor, Mineralocorticoid receptor, Cytochrome P450 17A1, Carboxylesterase 2, 11β-hydroxysteroid dehydrogenase 1
(16)	Cytochrome P450 19A1, Melatonin receptor 1A, Melatonin receptor 1B, Epoxide Hydrolase 1, Epoxide Hydratase, Estradiol 17-β-dehydrogenase 2.
(17)	Cytochrome P450 19A1, Cytochrome P450 17A1, 11β-hydroxysteroid dehydrogenase 1, Nitric oxide synthase inducible, Steroid 5α-reductase 2, Androgen receptor, Progesterone receptor.
(18)	Protein tyrosine phosphatase 1B, Cytochrome P450 19A1, Cytochrome P450 17A1, 11β-hydroxysteroid dehydrogenase 1, Acetyl-CoA Carboxylase 2, Vascular endothelial growth factor receptor 2.
(19)	Cytochrome P450 19A1, Cytochrome P450 17A1, 11β-hydroxysteroid dehydrogenase 1, Acetylcholinesterase, Glucocorticoid receptor, Mineralocorticoid receptor.
(20)	Cytochrome P450 19A1, Cytochrome P450 17A1, 11β-hydroxysteroid dehydrogenase 1, Androgen receptor, Glucocorticoid receptor, Progesterone receptor.
(21)	Cytochrome P450 19A1, Cytochrome P450 17A1, Androgen receptor, Glucocorticoid receptor, Progesterone receptor, Nitric oxide synthase inducible.
(22)	Cytochrome P450 19A1, Cytochrome P450 17A1, 11β-hydroxysteroid dehydrogenase 1, Glucocorticoid receptor, Progesterone receptor, Nitric oxide synthase inducible.
(23)	Cytochrome P450 19A1, Cytochrome P450 17A1, 11β-hydroxysteroid dehydrogenase 1, Glucocorticoid receptor, Progesterone receptor, Testis-specific androgen-binding protein, Nitric oxide synthase inducible.
(24)	Androgen receptor, progesterone receptor, Glucocorticoid receptor, Cytochrome P450 2C19, Mineralocorticoid receptor, Testis-specific androgen-binding protein, Steroid 5α-reductase 2.
(25)	Cytochrome P450 19A1, Cytochrome P450 17A1, Mineralocorticoid receptor, Progesterone receptor, Androgen receptor, Epoxide hydratase, Endoplasmin.
(26)	Cytochrome P450 17A1, Mineralocorticoid receptor, Progesterone receptor, Androgen receptor, Epoxide hydratase, Cytochrome P450 19A1.
(27)	Orexin receptor 1, Orexin receptor 2, Cathepsin S, Calciun sensing receptor, Cyclin-dependent Kinase 1, Phosphodiesterase 10A, Vasopressin V2 receptor.
(28)	Cytochrome P450 19A1, Cytochrome P450 17A1, Mineralocorticoid receptor, Progesterone receptor, Androgen receptor, Estrogen receptor α, Estrogen receptor β.
(29)	Cytochrome P450 19A1, Protein tyrosine phosphatase 1B, Thromboxane A2 receptor, Progesterone receptor, Androgen receptor, Mineralocorticoid receptor.
(30)	Progesterone receptor, Androgen receptor, Estrogen receptor β, Cytochrome P450 2C19, Mineralocorticoid receptor, Glucocorticoid receptor.
(31)	Testis-specific androgen-binding protein, Glucose-6-Phosphate 1-dehydrogenase, Corticosteroid binding globulin, Cytochrome P450 19A1, Estrogen receptor β, Androgen receptor, Glucocorticoid receptor, Cytochrome P450 17A1.

**Table 2 cimb-47-01059-t002:** Dimethyl sulfoxide (DMSO), Insulin (50 U/kg weight) single doses, Steroid 12 (11 mg/kg weight).

	Group 1	Group 2	Group 3	Group 4
Treatment	Basal (No treatment)	Insulin + DMSO	Insulin + Steroid 12 (single doses)	Insulin + Steroid 12 (four doses)

**Table 3 cimb-47-01059-t003:** Novel synthetic steroidal structures and Kinetic assay.

Steroids	5α-Reductase ± SD	Km ± SD	Vmax ± SD
(30) Spironolactone C_24_H_32_O_4_S MW = 416.57 g/mol	0.305 ± 0.006	0.0309 ± 0.006	0.304 ± 0.007
(16) 1,4,6-tripregnen-20-one C_21_H_28_O MW = 296 g/mol	0.739 ± 0.01	0.378 ± 0.03	0.767 ± 0.01
(22) 3β-acetoxy-5-pregnen-17α-hexanoiloxy-20-one C_29_H_44_O_5_ MW = 472 g/mol	0.615 ± 0.16	0.288 ± 0.27	0.607 ± 0.16
(24) 4-chloro-5-pregnen-17α-etiloixy-3,20-dione C_23_H_31_ClO_4_ MW = 406.5 g/mol	0.369 ± 0.002	0.010 ± 0.008	0.370 ± 0.003
(26) 3α,4α-epoxy-17α-hexyl-5-pregnen-20-one C_27_H_42_O_2_ MW = 398 g/mol	0.406 ± 0.008	0.063 ± 0.009	0.409 ± 0.006
(27) 3α,4α-epoxy-17α-phenylacetyl-5-pregnen-20-one C_29_H_36_O_3_ MW = 432 g/mol	0.434 ± 0.007	0.075 ± 0.01	0.434 ± 0.007
(29) 3α,4α-epoxy-17α-valeryl-5-pregnen-20-one C_26_H_38_O_4_ MW = 414 g/mol	0.390 ± 0.008	0.020 ± 0.01	0.392 ± 0.008
(13) 16α-methyl-6-pregnen-17α-benzoiloxy-20-one C_29_H_36_O_4_ MW = 448 g/mol	0.381 ± 0.01	0.142 ± 0.014	0.382 ± 0.01

**Table 4 cimb-47-01059-t004:** Blood triglycerides and glucose levels in rats treated with novel steroid **12** and insulin. Mean values ± S.D.

Experimental Groups	Triglycerides mg/dL	Glucose mg/dL
Basal	128.6 ± 29	159.2 ± 10 *
DMSO + Insulin	107.4 ± 4	31.0 ± 3
Steroid 12 (SD) + Insulin	94.0 ± 31	31.8 ± 8
Steroid 12 (RD) + Insulin	116.6 ± 9	35.6 ± 5

DMSO = Dimethyl Sulfoxide, SD = Single Dose, RD = Repeated Doses. Insulin administrated in experimental groups significantly (* *p* < 0.05) decreased the glucose levels in blood, inducing hypoglycemia, although the triglycerides decreased partially.

## Data Availability

The data presented in this study are available on request from the corresponding author due to is one of his functions.

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
