# Peer review of "Curr. Issues Mol. Biol.2025, 47(12), 1059;https://doi.org/10.3390/cimb47121059"

_cimb, 2025, doi:10.3390/cimb47121059_

Round 1

Reviewer 1 Report

Comments and Suggestions for Authors

The manuscript examines novel steroid derivatives with potential antiandrogenic activity, integrating computational target prediction and biological assays. Combining in silico and in vivo approaches is important for drug discovery. However, the manuscript requires improved clarity, methodological rigor, and data presentation to meet publication standards. The following points need to be reviewed and corrected: Clarity of objectives and scientific coherence. The abstract currently blends elements of molecular target prediction, biological evaluation, and potential implications in COVID-19, but a more explicit connection between them would enhance clarity. It would be helpful to reformulate the research goals, clearly state the hypotheses, and define the scope of the study. The authors should further review the methodological description. Additional details on the artificial intelligence tools and algorithms used, such as the version of SwissTargetPrediction, validation process, and performance metrics, would strengthen the manuscript by enhancing transparency, reproducibility, and reliability of computational results. Specifying algorithm parameters, data sources, and validation metrics enables assessment of predictive performance, minimizes bias, and supports the correlation between in silico predictions and biological assays of the novel steroid derivatives. The biological evaluation section would be improved by providing additional information, the sample size may not be fully representative, control groups, statistical analyses, and ethical approval. Similarly, including more details on the chemical synthesis and structural characterization (NMR, HRMS, IR, PXRD) of the steroid derivatives (purity and yields).   Improving the results and data presentation would strengthen the manuscript. The reported findings (neuroprotection, 5α-reductase inhibition) would be supported by quantitative data or figures. Providing statistical significance and comparisons with reference compounds would add value. Additionally, a clearer analysis and discussion of the relationship between computational predictions and biological results would contribute to scientific understanding. The discussion and conclusions section could be expanded. Providing a more detailed interpretation of the findings in the context of existing literature on synthetic antiandrogens would be valuable. If referencing COVID-19 progression, offering supporting experimental data or clarifying the relevance would improve this section. Other issues:

  • Clarify the chemical names of the synthesized compounds.
  • The rationale for linking the study to COVID-19 progression is unclear. The manuscript does not present data supporting this connection, and this linkage needs explicit justification or should be omitted to avoid confusing the study’s focus.

The topic has scientific value and may aid the rational design of steroidal antiandrogens with neuroprotective effects. Revisions to improve methodological transparency, data presentation, and scientific communication would strengthen the manuscript.

Comments on the Quality of English Language

The manuscript needs substantial English editing to correct several grammatical and stylistic errors (e.g., "theraphy" for "therapy", "demostrated" for "demonstrated") and improve readability. A thorough revision by a native English speaker or professional editor is recommended.

Author Response

Reviewer 1

Comments and Suggestions for Authors:

The manuscript continues to have serious deficiencies in its figures, as they are not numbered, making it impossible to cite them within the text. Additionally, the presentation of chemical structures is careless, with inconsistent sizes, styles, and fonts, which should be standardized. Furthermore, the statement regarding the relationship between severe COVID-19 cases and androgens remains inconclusive and insufficiently substantiated. These issues are interrelated and hinder the clarity and credibility of the manuscript's findings. The authors now state they have synthesized all the steroidal derivatives. It would be valuable to explain the rationale behind selecting only one compound for biological testing.

It was made in new version.

More men died of coronavirus disease 2019 (COVID-19) than women, suggesting estrogens may protect women (Abdulmaged M Traish. Sex steroids and COVID-19 mortality in women. Trends Endocrinol Metab. 2021 Aug;32(8):533-536. doi: 10.1016/j.tem.2021.04.006. Epub 2021 Apr 19.).

Emerging reports have shown the benefits of steroids in hospitalized COVID-19 patients as life-saving drugs. Dexamethasone as steroid, during severe COVID-19 affect circulating neutrophils, alter IFNactive neutrophils, downregulate interferon-stimulated genes and activate IL-1R2+ neutrophils, and expand immunosuppressive immature neutrophils and remodel cellular interactions by changing neutrophils from information receivers into information providers. Male patients have higher proportions of IFNactive neutrophils and preferential steroid-induced immature neutrophil expansion, potentially affecting outcomes (Sarthak Sinha, Nicole L Rosin, Rohit Arora, Elodie Labit, Arzina Jaffer, Leslie Cao, Raquel Farias, Angela P Nguyen, Luiz G N de Almeida, Antoine Dufour, Amy Bromley, Braedon McDonald, Mark R Gillrie, Marvin J Fritzler, Bryan G Yipp, Jeff Biernaskie. Dexamethasone modulates immature neutrophils and interferon programming in severe COVID-19. Nat Med. 2022 Jan;28(1):201-211. doi: 10.1038/s41591-021-01576-3. Epub 2021 Nov 15.), and recovery.

Comments on the Quality of English Language:

The manuscript needs substantial English editing to correct several grammatical and stylistic errors (e.g., "theraphy" for "therapy", "demostrated" for "demonstrated") and improve readability. A thorough revision by a native English speaker or professional editor is recommended.

 It was made in new version

Reviewer 2 Report

Comments and Suggestions for Authors

This paper contains some interesting work assessing the potential of synthetic steroids as antiandrogens. In particular, the authors have used AI to predict targets of these novel compounds which could be used to determine suitability for use in patients. They have also tested the effect of one particular drug (Steroid 12) in a mouse model.

Major points to address:

  • The authors need to be clearer upfront on their experimental model. The use of Insulin to induce a hypoglycemia is mentioned in the abstract and discussion, but needs to be more explicit in the introduction and methods.
  • The authors mention treatment of COVID-19 and hormonal therapies a number of times. While there may be some application from the target prediction data, it is harder to justify the use of Steroid 12 for these applications based on the hypoglycemia model.
  • The discussion of results needs to be expanded. There is some interpretation of results in the discussion section, but it could go further.
  • There appears to be no statistical tests performed for data in Tables 2 & 3, or at least not discussed.
  • I'm struggling to agree with your interpretation of the data. For data in Figures 2-6, there is a clear effect of Insulin compared to Basal. However, the administration of steroid 12 appears to have no significant effect compared to the DMSO + Insulin group.
  • Therefore, I would argue that the claim made by the authors starting on Line 410, "The use of steroids in the treatment of the young animals with hypoglycaemia induced, increased glutathione (GSH) in cortex, hemispheres and medulla oblongata, decreasing the 5-HIAA in the same tissues", does not appear to be true given the evidence presented.

Minor suggestions/corrections:

  • Line 72 - Suggest to give examples of physiological function of sulfonated steroids.
  • Line 86 - "a specific drug", may want to rephrase. In pharmacology, specificity is normally the ability of a drug to bind to a single, specific target.
  • Line 90 - "overall testosterone production of the adrenal cortex", what about in the testes?
  • Line 174 - "a peptide antagonist", include an example.
  • Table 1 - The "Steroid or Structure" is blank between numbers 19 and 20. Should this be with compound 19?
  • Line 296, 308, 319 - Consider including concentration ranges used for constructing standard curve.
Comments on the Quality of English Language

There are some instances where English could be improved for clarity:

  • Line 17 - "Two steroids derived from cholesterol 17 pathways are testosterone and progesterone, as antiandrogen – estrogen, respectively" needs to be clearer. Unsure what estrogen is referring to here.
  • Line 21 - "In COVID-19 progression sex hormones regulate", needs a comma after progression
  • Line 27 - "and was made the pharmacological 27 evaluation", needs to be clearer.
  • Line 32 - "theraphy", typo
  • Line 33 - "Respect", should this be respectively?
  • Line 65 - "It hormones", should be "These hormones"
  • Line 77 - "antiandrogens as dehydropregnenolone" should use "such as"
  • Line 130 - "nd", should be "and"?
  • Line 135 - "They are unique because they do not counteract the effects or production of other androgens other than DHT", suggest replacing "they" with "Antiandrogens" to avoid ambiguity.
  • Line 141 - "inhibitors as finasteride" should be "such as"
  • Line 167 - "and these exogenous hormones", suggest this is a new sentence.
  • Line 209 - "The model predicting of target prediction", could be reworded
  • Line 222 - "antagonists due mainly 222 cyclopentano-perhydrophenanthrene structure", suggest replacing "due mainly" with "based on".
  • Line 229 - "is favorable insert" should this be "is favorable to insert"?
  • Line 264 - "formed by first group", should be "formed the first group"
  • Line 265 - "formed by second group", should be "formed the second group"
  • Line 281 & 344 - "Buffer phosphate solution", should be "phosphate buffer solution"
  • Line 352 - "cuadruplicate" should be "quadruplicate"

Author Response

Reviewer 2

Comments and Suggestions for Authors:

I thank the authors for the revised version of the manuscript. I'm satisfied that all comments have been addressed.

THANKS

Reviewer 3 Report

Comments and Suggestions for Authors

The study presents an interesting attempt to combine computational target prediction with biological validation of novel steroid derivatives derived from 16-dehydropregnenolone acetate. The work links antiandrogenic activity, oxidative stress biomarkers, and enzyme kinetics (5α-reductase inhibition), which are all relevant to neuroprotection and potential COVID-19-related hormonal mechanisms.
However, the manuscript requires major revision to enhance clarity, organization, scientific rigor, and language quality. Several concepts are insufficiently explained, and grammatical inconsistencies reduce readability. Additionally, the results and conclusions need stronger justification and clearer data presentation. The manuscript could be considered for publication in CIMB after substantial revision.

Major comments:

  1. I recommend modifying the title. It’s a little bit ambiguous and grammatically inconsistent. The use of a period (.) in the middle makes it look like two unrelated statements. It would read more professionally as a single coherent sentence. (for example, you can change it to Novel Synthetic Steroid Drugs (or derivatives?): Target Prediction and Biological Evaluation of Antiandrogenic Activity)
  2. I recommend revising the abstract for greater clarity and accuracy. For instance, the opening sentences are confusing and contain grammatical errors (e.g., the phrase “as antiandrogen – estrogen, respectively” is unclear and should be reworded).
  3. The Introduction section exhibits a high similarity with previously published sources. The authors are encouraged to rephrase this section in their own words to improve originality while continuing to accurately cite relevant literature.
  4. Throughout the manuscript, the authors alternate between “anti-androgen” and “antiandrogen.” 5α-reductase and 5alpha-reductase. Please ensure consistent terminology is used across the entire text.
  5. The description and presentation of the chemical structures are unclear. The numbering of the structures is not sequential, and some structures lack indicated chirality. It is recommended to revise the figures to improve clarity and readability. For example, use “Testosterone (1)” instead of “1. Testosterone.”
  1. The use of abbreviations is inconsistent throughout the manuscript. For example, in lines 132 and 135, the authors use the abbreviation DHT and later refer to dihydrotestosterone in full. Please ensure that each abbreviation is defined the first time it appears in the text (e.g., dihydrotestosterone (DHT)) and that the abbreviation is used consistently thereafter. The same issue occurs with other terms, such as ROS.
  1. Please include an illustration or labeling in one representative structure to clearly indicate rings A-D of the steroid nucleus. This will help readers unfamiliar with steroid structural conventions to easily identify and follow your discussion of structural modifications.
  2. It is unclear why steroid 12 was specifically selected for most of the biological activity studies, while the other steroid derivatives were not. Although this may be implied in the manuscript, it is not stated clearly. I recommend adding a short paragraph at the end of the Introduction to explicitly explain the rationale for choosing this compound.
  3. I recommend adding explanatory text to accompany the figures in the Results. In the current version, the authors present only graphs without any descriptive interpretation. Each figure should be supported by a concise explanation summarizing the main findings and their relevance. Consider organizing the Results section into subsections to improve clarity and flow. Additionally, in the kinetic study, please indicate whether a positive control was included and provide details regarding its use and results.
  4. In the Discussion section, the authors should further elaborate on and interpret these findings, explaining their significance in the context of the study’s objectives and existing literature.
  5. Please indicate the source of all drugs or steroid derivatives used in the study. Specify whether they were purchased commercially (including supplier name and country) or synthesized in-house, and provide relevant details in the Materials and Methods

Round 2

Reviewer 1 Report

Comments and Suggestions for Authors

The manuscript continues to have serious deficiencies in its figures, as they are not numbered, making it impossible to cite them within the text. Additionally, the presentation of chemical structures is careless, with inconsistent sizes, styles, and fonts, which should be standardized. Furthermore, the statement regarding the relationship between severe COVID-19 cases and androgens remains inconclusive and insufficiently substantiated. These issues are interrelated and hinder the clarity and credibility of the manuscript's findings. The authors now state they have synthesized all the steroidal derivatives. It would be valuable to explain the rationale behind selecting only one compound for biological testing.

Comments on the Quality of English Language

The manuscript needs substantial English editing to correct several grammatical and stylistic errors (e.g., "theraphy" for "therapy", "demostrated" for "demonstrated") and improve readability. A thorough revision by a native English speaker or professional editor is recommended.

Author Response

The manuscript continues to have serious deficiencies in its figures, as they are not numbered, making it impossible to cite them within the text. Additionally, the presentation of chemical structures is careless, with inconsistent sizes, styles, and fonts, which should be standardized. Furthermore, the statement regarding the relationship between severe COVID-19 cases and androgens remains inconclusive and insufficiently substantiated. These issues are interrelated and hinder the clarity and credibility of the manuscript's findings. The authors now state they have synthesized all the steroidal derivatives. It would be valuable to explain the rationale behind selecting only one compound for biological testing.

It was made in new version.

More men died of coronavirus disease 2019 (COVID-19) than women, suggesting estrogens may protect women (Abdulmaged M Traish. Sex steroids and COVID-19 mortality in women. Trends Endocrinol Metab. 2021 Aug;32(8):533-536. doi: 10.1016/j.tem.2021.04.006. Epub 2021 Apr 19.).

Emerging reports have shown the benefits of steroids in hospitalized COVID-19 patients as life-saving drugs. Dexamethasone as steroid, during severe COVID-19 affect circulating neutrophils, alter IFNactive neutrophils, downregulate interferon-stimulated genes and activate IL-1R2+ neutrophils, and expand immunosuppressive immature neutrophils and remodel cellular interactions by changing neutrophils from information receivers into information providers. Male patients have higher proportions of IFNactive neutrophils and preferential steroid-induced immature neutrophil expansion, potentially affecting outcomes (Sarthak Sinha, Nicole L Rosin, Rohit Arora, Elodie Labit, Arzina Jaffer, Leslie Cao, Raquel Farias, Angela P Nguyen, Luiz G N de Almeida, Antoine Dufour, Amy Bromley, Braedon McDonald, Mark R Gillrie, Marvin J Fritzler, Bryan G Yipp, Jeff Biernaskie. Dexamethasone modulates immature neutrophils and interferon programming in severe COVID-19. Nat Med. 2022 Jan;28(1):201-211. doi: 10.1038/s41591-021-01576-3. Epub 2021 Nov 15.), and recovery.

The manuscript needs substantial English editing to correct several grammatical and stylistic errors (e.g., "theraphy" for "therapy", "demostrated" for "demonstrated") and improve readability. A thorough revision by a native English speaker or professional editor is recommended.

 It was made in new version.

Reviewer 2 Report

Comments and Suggestions for Authors

I thank the authors for the revised version of the manuscript. I'm satisfied that all comments have been addressed.

Author Response

Thanks.

Reviewer 3 Report

Comments and Suggestions for Authors

The authors have addressed most of the previous comments appropriately. However, a few issues still need to be clarified and revised for improved clarity.

It is not clearly stated whether the compounds were synthesized in-house by the authors. If this is the case, please include the NMR spectra and detailed characterization methods in the Supporting Information.

Additionally, the reaction scheme requires correction, as it is currently unclear and poorly formatted. Please revise it for better clarity and presentation.

Please also ensure that brackets ( ) are consistently placed before and after each compound number throughout the manuscript.

Furthermore, it is unclear why the authors refer to "Figure 32" while the numbering in the Results section begins from "Figure 2." The figure numbering should be checked and corrected for consistency, and the numbering of figures should be clearly separated from the numbering of compounds, as these are two distinct categories.

good luck 

Round 3

Reviewer 3 Report

Comments and Suggestions for Authors

I recommend including the experimental data for the preparation of the compounds in the Supporting Information.